# Lightweight Generative Adversarial Networks for Text-Guided Image Manipulation

**Bowen Li[1], Xiaojuan Qi[2], Philip H. S. Torr[1], Thomas Lukasiewicz[1]**
[1]University of Oxford, [2]University of Hong Kong
{bowen.li, thomas.lukasiewicz}@cs.ox.ac.uk
xjqi@eee.hku.hk, philip.torr@eng.ox.ac.uk

## Abstract

We propose a novel lightweight generative adversarial network for efficient image manipulation using natural language descriptions. To achieve this, a new word-level discriminator is proposed, which provides the generator with fine-grained training feedback at word-level, to facilitate training a lightweight generator that has a small number of parameters, but can still correctly focus on specific visual attributes of an image, and then edit them without affecting other contents that are not described in the text. Furthermore, thanks to the explicit training signal related to each word, the discriminator can also be simplified to have a lightweight structure. Compared with the state of the art, our method has a much smaller number of parameters, but still achieves a competitive manipulation performance. Extensive experimental results demonstrate that our method can better disentangle different visual attributes, then correctly map them to corresponding semantic words, and thus achieve a more accurate image modification using natural language descriptions.

## 1  Introduction

How to effectively edit a given image without tedious human operation is a challenging but meaningful task, which may potentially boost enormous applications in different areas, such as design, architecture, video games, and art. Recently, with the great progress on the development of deep learning, various applications in terms of image manipulation have been developed, including style transfer [5, 6, 9, 11], image colourisation [2, 13, 32], and image translation [3, 10, 20, 21, 26, 28].

Differently from above works, the goal of this paper is to provide a more user-friendly method, which can automatically edit a given image by simply using natural language descriptions. In particular, we aim to semantically modify parts of an image (e.g., colour, texture, and global style) according to user-provided text descriptions, where the descriptions contain desired visual attributes that the modified image should have. Meanwhile, the modified result should preserve text-irrelevant contents of the original image that are not required by the text.

Currently, only few studies [4, 15, 19] work on this task. Methods introduced in [4, 19] both fail to effectively modify text-required attributes, and results are also far from satisfactory (see Fig. 1). Recently, Li et al. [15] proposed a new multi-stage network, which is able to produce more realistic images. However, as the model [15] is based on a multi-stage framework with multiple pairs of generator and discriminator, it very likely requires a large memory and needs a lot of time for training and inference, which is less practical to memory-limited devices, such as mobile phones.

This motivates us to investigate a lightweight architecture of the network. However, simply reducing the number of stages and parameters in the model [15] cannot achieve a satisfactory result, shown in Fig. 1 (e). As we can see, compared with the original ManiGAN (d), the image quality of synthetic results drops significantly as both images are obviously blurred. Also, the manipulation ability of this

Top: A **red** bird has **black eye rings** and **black wings**, with a **red crown** and a **red belly**.
Bottom: Vase, **red flowers**.

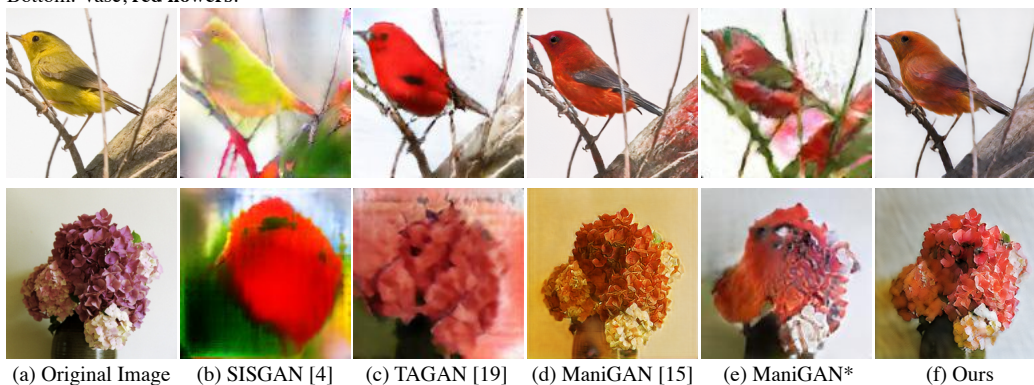

(a) Original Image     (b) SISGAN [4]     (c) TAGAN [19]     (d) ManiGAN [15]     (e) ManiGAN*     (f) Ours

Figure 1: Examples of image manipulation using natural language descriptions. Our method has a lightweight architecture, but can still allow the input images to be manipulated accurately matching the descriptions. "ManiGAN*" denotes we reduce the number of stages and parameters in the model.

lightweight ManiGAN becomes worse, because more regions (e.g., sky and branches) are coloured red. After a close investigation, we find that the discriminator used in the model [15] fails to provide the generator with fine-grained training feedback related to each word, because it only calculates the similarity between the whole text and image features. Due to the coarse supervisory feedback from discriminators, the network needs to have a heavy structure with a large number of parameters, to build an accurate relation between visual attributes and corresponding semantic words to achieve an effective manipulation, which greatly impedes the construction of a lightweight architecture. Additionally, even in the original ManiGAN, this poor training feedback prevents the model from completely disentangling different visual attributes, causing an incorrect mapping between attributes and corresponding semantic words. Thus, some text-irrelevant contents are changed. For example, as shown in Fig. 1 (d), the branch is coloured red, and the background of the vase is modified as well.

Based on this, we propose a new word-level discriminator along with explicit word-level supervisory labels, which can provide the generator with detailed training feedback related to each word, to facilitate training a lightweight generator that has a small number of parameters but can still effectively disentangle different visual attributes, and then correctly map them to the corresponding semantic words. Besides, thanks to our powerful discriminator to provide explicit word-level training feedback, our network can be simplified to have only a generator network and a discriminator network, and we can even further reduce the number of parameters in the model without sacrificing much image quality, which is more friendly to memory-limited devices.

To this end, we evaluate our model on the CUB bird [27] and more complicated COCO [17] datasets, which demonstrates that our method can accurately modify the given image using natural language descriptions with great efficiency. Also, extensive experiments on both datasets show the superiority of our method, in terms of both visual fidelity and efficiency. The code will be available at https://github.com/mrlibw/Lightweight-Manipulation.

## 2 Related Work

**Text-guided image manipulation** has drawn much attention, due to its great potential in enabling a more general and easier tool for users, where users can simply edit an image using natural language descriptions. Dong et al. [4] proposed an encoder-decoder architecture to modify an image matching a given text. Nam et al. [19] disentangled different visual attributes by introducing a text-adaptive discriminator, which can provide finer training feedback to the generator. Li et al. [15] proposed a multi-stage network with a novel text-image combination module to produce high-quality results. However, modified images produced by both methods [4, 19] are far from satisfactory, and the method [15] fails to completely disentangle different attributes and is limited on the efficiency as well.

**Text-to-image generation** focuses on generating images from texts. Mansimov et al. [18] iteratively draws patches on a canvas, while attending to the relevant words in the description. Reed et al. [22, 23]

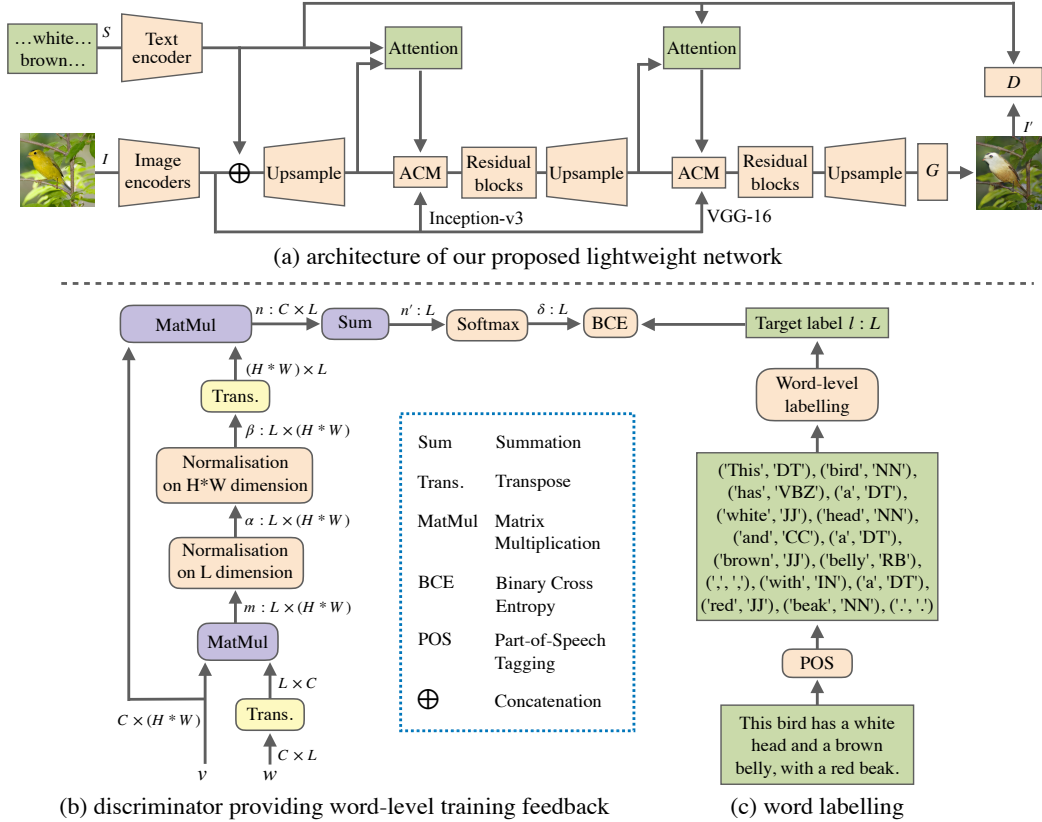

(a) architecture of our proposed lightweight network

(b) discriminator providing word-level training feedback

(c) word labelling

Figure 2: Top: the architecture of our model. Inception-v3 and VGG-16 denote image features are extracted by the corresponding image encoder. Bottom: the design of our word-level discriminator.

applied GANs [7] to generate synthetic images matching the given texts. Zhang et al. [30, 31] stacked multiple GANs to refine the generated images progressively. Xu et al. [29] and Li et al. [14] introduced attention mechanisms to explore word-level information. However, all above methods aim to generate new images from given texts instead of editing a given image using text descriptions.

**Image-to-image translation** is also related to our work. Zhu et al. [33] proposed to explore latent vectors to manipulate the attributes of an object. Wang et al. [28] introduced a multi-scale architecture to produce high-resolution images. Park et al. [20] applied affine translations to avoid information loss caused by normalisation. Li et at. [16] used text descriptions to control the image translation. However, all these methods aim to generate a realistic image from a semantic mask, instead of modifying a real image using cross-domain natural language descriptions.

**Word-level discriminator.** In order to provide a finer training feedback to the generator, different word-level discriminators have been studied. The text-adaptive discriminator [19] utilised a word-level text-image matching score as supervision to enable detailed feedback. However, a global pooling layer was adopted in the discriminator to generate image features, which causes the loss of important spatial information. To avoid the loss problem, Li et al. [14] incorporated word-level spatial attention into the discriminator. However, the cosine similarity is directly applied on the whole text and image features, which actually fails to fully explore the word-level information in text features, and thus leads to a coarse training feedback.

## 3   Lightweight Generative Adversarial Networks for Image Manipulation

Given an image $I$ and a text description $S$, we aim to modify the input image $I$ according to the description $S$, to produce a new modified image $I'$ that contains all new visual attributes described in the text $S$, while preserving other text-irrelevant contents. Meanwhile, the model should be small

and efficient enough to memory-limited devices. To achieve this, we propose a novel word-level discriminator, which helps to achieve a lightweight architecture. We elaborate our model as follow.

## 3.1 Word-Level Discriminator

To facilitate training an effective lightweight generator that has a small number of parameters but can still achieve a satisfactory manipulation performance, the discriminator should provide the generator with fine-grained training feedback in terms of each word in the given sentence. Although there are several studies working on this, the text-adaptive discriminator in [19] loses important image spatial information, due to the global average pooling layer, and the discriminator proposed in [14] ignores the word information, within text features caused by the direct implementation of the cosine similarity between the whole text and image features. To address this, we propose a novel word-level discriminator to fully explore the word-level information contained in text features, and so build an effective independent relation between each visual attribute and the corresponding semantic word.

### 3.1.1 Word Labelling

In order to fully explore the word-level information, we first label each word in the description using part-of-speech tagging [1], which marks up each word to a particular part of speech (e.g., noun, verb, adjective, etc.), based on its definition and context. Then, we create a target label $l$ having the same length as the number of words in the sentence, where each **noun** and **adjective** is set to 1 and others to 0. This target label $l$ is used to calculate the loss function (Eq. 3) as word-level training feedback.

**Why do we only keep nouns and adjectives?** Given an image, what we can observe mainly are objects and their corresponding visual attributes, which correspond to nouns and adjectives in the description, respectively. Also, we aim to disentangle different visual attributes existing in the image and then build an accurate relation between visual attributes and corresponding semantic words. That is, the model only needs to identify the category of objects and their attributes, and then maps them to corresponding semantic words (nouns and adjectives) to build a correct relation. Moreover, there is no need to keep potential logical relation (e.g., relative position) contained in the pronoun or verb, as this information has already been kept in the input image.

### 3.1.2 Discriminator Providing Word-Level Training Feedback

Our novel word-level discriminator takes two inputs: (1) the image features $v$ and $v'$ encoded by the pre-trained Inception-v3 [25] image encoder from the modified images $I$ and $I'$, respectively, and (2) the word features $w \in \mathbb{R}^{C \times L}$ encoded by the pre-trained text encoder [14, 29] from the given text description $S$, where $C$ is the feature dimension, and $L$ denotes the number of words in $S$.

In the following, for simplicity, we use $v \in \mathbb{R}^{C \times (H*W)}$ to represent both real image features and synthetic image features. First, we compute the word-region correlation matrix $m \in \mathbb{R}^{L \times (H*W)}$ via $m = w^T v$, which contains the information for all pairs of words in the description $S$ and regions of the image $I$ (or $I'$). Then, we normalise the word-region correlation matrix on both $L$ and $H * W$ dimensions sequentially using the following equations:

$$\alpha_{i,j} = \frac{\exp(m_{i,j})}{\sum_{k=0}^{L-1} \exp(m_{k,j})}, \quad \beta_{i,j} = \frac{\exp(\alpha_{i,j})}{\sum_{k=0}^{(H*W)-1} \exp(\alpha_{i,k})}, \tag{1}$$

where $\beta_{i,j}$ represents the probability value that the $i^{th}$ word is relevant to the $j^{th}$ region of the image. Then, the word-weighted image features $n \in \mathbb{R}^{C \times L}$ can be derived via $n = v\beta^T$, where $n_{i,j}$ represents the sum of image features at the $i^{th}$ channel, and each spatial region at the $i^{th}$ channel is weighted by the $j^{th}$ word in the description. Next, we further sum the word-weighted image features $n$ at the $C$ dimension to get $n' \in \mathbb{R}^L$, and normalise $n'$ by the softmax function:

$$\delta_i = \frac{\exp(n_i')}{\sum_{k=0}^{L-1} \exp(n_k')}, \tag{2}$$

where $\delta_i$ reflects the correlation between the $i^{th}$ word and the whole image. That is, it represents the chance that $i^{th}$-word-related visual attributes exist in the image. Thus, the word-level feedback used to train the generator can be calculated using Eq. 3:

$$\mathcal{L}_{\text{word}}(I, S) = \text{BCE}(\delta, l), \tag{3}$$

where BCE denotes binary cross-entropy, and $l$ is the target label (see Sec. 3.1.1). By doing this, the discriminator provides the generator with explicit fine-grained training feedback at word-level, helps to disentangle different words in the sentence and visual attributes in the image, and enables an independent modification of each image regions according to the text descriptions. Note that there are no trainable parameters in our word-level discriminator, and the implementation of it does not affect the context of the general description, as the objective functions (Eqs. 5 and 6) have included the full sentence ($S$) as conditional adversarial losses to convey a rich context of text.

## 3.2 Generator of the Lightweight Architecture

Thanks to our word-level discriminator providing fine-grained training feedback related to each word, we are able to build a lightweight generator with a simple structure, as shown in Fig. 2. The generator consists of a text encoder, image encoders providing two different image features, and a series of upsampling and residual blocks, where the text encoder is a pre-trained bidirectional RNN [14, 29], and the image encoders are pre-trained Inception-v3 [25] and VGG-16 [24] networks, respectively.

First, the generator encodes the input image $I$ into image features using the Inception-v3 network, and then concatenates them with text features, encoded by the text encoder from the description $S$. Next, the image-text features are fed into a series of upsampling and residual blocks, followed by an image generation network to produce the modified result with the desired resolution.

We adopt conditioning augmentation [30] to smooth text representations, and the text-image affine combination module (ACM) from [15] to fuse text and image features. Note that for the last ACM, we use the VGG-16 network to extract image features, which contains more content details, helping rectify inappropriate attributes and complete missing contents. Besides, spatial attention [29] and channel-wise attention [14] are applied. Following [14], perceptual loss [11] is adopted to reduce the randomness involved in the generation process and to help preserve text-irrelevant contents:

$$\mathcal{L}_{\text{per}}(I', I) = \frac{1}{C_i H_i W_i} \|\phi_i(I') - \phi_i(I)\|_2^2, \tag{4}$$

where $I$ is the input image, $I'$ is the modified result, $\phi_i(I)$ is the activation of the $i^{th}$ layer of a pre-trained VGG-16 network, $C_i$ is the number of channels, and $H_i$ and $W_i$ are the height and width of the feature map, respectively.

**Why do we use two different image encoders?** Image features extracted by the deeper Inception-v3 network is more semantic, where these features only contain basic information of the input image (e.g., the layout, the category, and the rough shape of objects), and are easier to interact with corresponding semantic words to achieve an effective manipulation. Thus, we use this coarse image features in the beginning and middle of the network, enabling a better manipulation ability. On the contrary, image features extracted by the shallower VGG-16 network contain too many content details, where these details may enforce the generator to simply reconstruct the input image and restrict the generation of new attributes required in the text description. Based on this, we use these informative details in the end of the model to rectify modified text-irrelevant attributes and complete missing contents.

## 3.3 Objective Functions

Only a pair of generator and discriminator exists in our model, and we train the generator and the discriminator alternatively by minimising both the generator loss $\mathcal{L}_G$ and the discriminator loss $\mathcal{L}_D$. Following [15], we use paired data $(I, S) \rightarrow I$ to train our model, where S is the text description matching the image $I$.

**Generator Objective.** The complete generator objective (Eq. 5) consists of unconditional and conditional adversarial losses, a perceptual loss $\mathcal{L}_{\text{per}}$ (Eq. 4), a word-level training loss $\mathcal{L}_{\text{word}}$ (Eq. 3), and a text-image matching loss $\mathcal{L}_{\text{DAMSM}}$ [29].

$$\mathcal{L}_G = \underbrace{-\frac{1}{2}E_{I' \sim P_G}\left[\log(D(I'))\right]}_{\text{unconditional adversarial loss}} \underbrace{-\frac{1}{2}E_{I' \sim P_G}\left[\log(D(I', S))\right]}_{\text{conditional adversarial loss}} \tag{5}$$
$$+ \lambda_1 \mathcal{L}_{\text{per}}(I', I) + \lambda_2 \mathcal{L}_{\text{word}}(I', S) + \lambda_3 \mathcal{L}_{\text{DAMSM}},$$

where $I$ is the real image sampled from the true image distribution $P_{\text{data}}$, $I'$ is the generated image sampled from the model distribution $P_G$, $\lambda_1$, $\lambda_2$, and $\lambda_3$ are hyperparameters controlling different losses, and $\mathcal{L}_{\text{DAMSM}}$ [29] measures the text-image matching score based on the cosine similarity.

Table 1: Quantitative comparison: Fréchet inception distance (FID), accuracy, and realism of the state of the art and our method on CUB and COCO. "w/o Dis." denotes without implementing proposed word-level discriminator. "w/ TAD" denotes using the text-adaptive discriminator [19] instead of our word-level discriminator. "w/ CD" denotes implementing the word-level discriminator introduced in ControlGAN [14]. For FID, lower is better; for accuracy and realism, higher is better.

| Method | | CUB | | | COCO | |
| | FID | Accuracy | Realism | FID | Accuracy | Realism |
|---|---|---|---|---|---|---|
| ManiGAN [15] | 9.75 | 34.06 | 42.18 | 25.08 | 22.03 | 32.47 |
| **Ours** | 8.02 | 65.94 | 57.82 | 12.39 | 77.97 | 67.53 |
| Ours w/o Dis. | 9.20 | - | - | 15.60 | - | - |
| Ours w/ TAD | 8.84 | - | - | 14.95 | - | - |
| Ours w/ CD | 8.72 | - | - | 13.68 | - | - |

Table 2: Quantitative comparison: number of parameters in generator (NoP-G) and discriminator (NoP-D), runtime per epoch (RPE), and inference time for generating 100 new modified images (IT) of the state of the art and our method on CUB and COCO. "Ours*" is the smallest number of parameters our model can have without significantly sacrificing image quality. For all evaluation matrices, lower is better. All methods are benchmarked on a single Quadro RTX 6000 GPU.

| Method | | CUB | | | COCO | | |
| | NoP-G | NoP-D | RPE (h) | IT (s) | NoP-G | NoP-D | RPE (h) | IT (s) |
|---|---|---|---|---|---|---|---|---|
| ManiGAN [15] | 41.1M | 169.4M | 0.138 | 2.470 | 53.3M | 377.6M | 1.992 | 4.485 |
| **Ours** | 18.5M | 71.8M | 0.089 | 0.425 | 30.1M | 160.7M | 1.475 | 0.718 |
| Ours* | 5.4M | 1.6M | 0.058 | 0.356 | 7.4M | 3.5M | 0.760 | 0.445 |

**Discriminator Objective.** The complete discriminator objective is defined as:

$$\mathcal{L}_D = \underbrace{-\frac{1}{2}E_{I \sim P_{\text{data}}}\left[\log(D(I))\right] - \frac{1}{2}E_{I' \sim P_G}\left[\log(1 - D(I'))\right]}_{\text{unconditional adversarial loss}}$$
$$\underbrace{-\frac{1}{2}E_{I \sim P_{\text{data}}}\left[\log(D(I, S))\right] - \frac{1}{2}E_{I' \sim P_G}\left[\log(1 - D(I', S))\right]}_{\text{conditional adversarial loss}} \quad (6)$$
$$+ \lambda_4(\mathcal{L}_{\text{word}}(I, S) + \mathcal{L}_{\text{word}}(I', S)),$$

where $\lambda_4$ is a hyperparameter. Note that the target label for this $\mathcal{L}_{\text{word}}(I', S)$ is set to 0 for all words.

## 4 Experiments

We evaluate our model on the CUB bird [27] and more complicated COCO [17] datasets, comparing with the current state of the art, ManiGAN [15], which also focuses on text-guided image manipulation. Results for the method are reproduced using the code released by the authors.

**Datasets.** The CUB bird [27] dataset contains 8,855 training images and 2,933 test images, where each image is along with 10 corresponding text descriptions. COCO [17] contains 82,783 training images and 40,504 validation images, where each image has 5 corresponding text descriptions. We preprocess these two datasets according to the method in [29].

**Implementation.** The scale of the output images is $256 \times 256$, but the size is adjustable to satisfy users' preferences. Similarly to [15], there is a trade-off between the generation of new attributes matching the text description and the preservation of text-irrelevant contents of the original image. Therefore, based on the manipulative precision (MP) [15], the whole model is trained 100 epochs

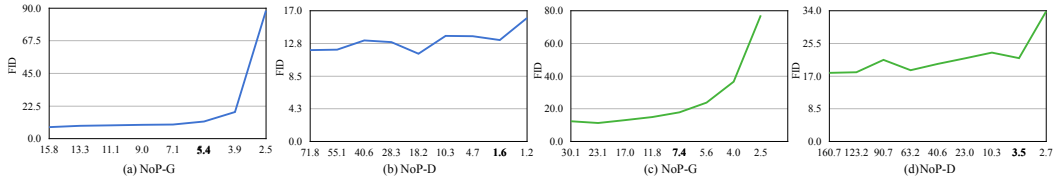

Figure 3: Fréchet inception distance (FID) at different numbers of parameters in the generator (or discriminator) when the number of parameters in the discriminator (or generator) is fixed. (a) and (b) are evaluated on the CUB dataset, and (c) and (d) are evaluated on the COCO dataset. In (a), the number of parameters in the discriminator is fixed at 71.8M; in (b), the number of parameters in the generator is fixed at 5.4M; in (c), the number of parameters in the discriminator is fixed at 160.7M; and in (d), the number of parameters in the generator is fixed at 7.4M.

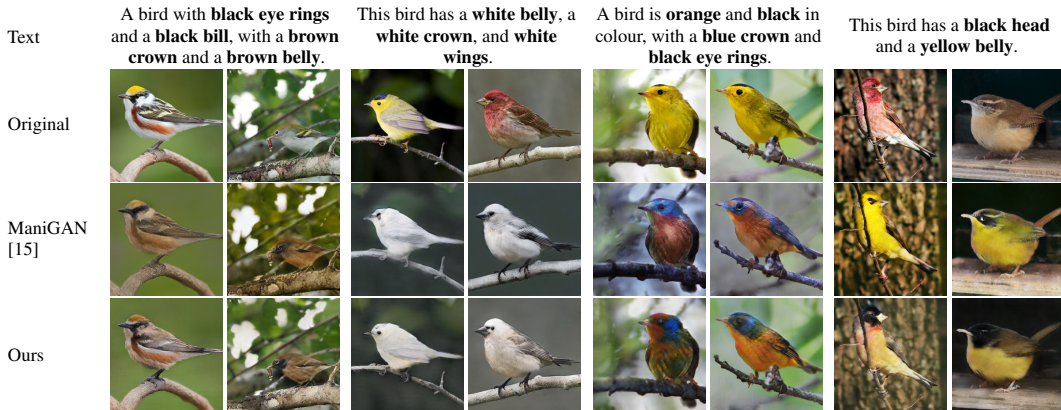

Figure 4: Qualitative comparison of two methods on the CUB bird dataset.

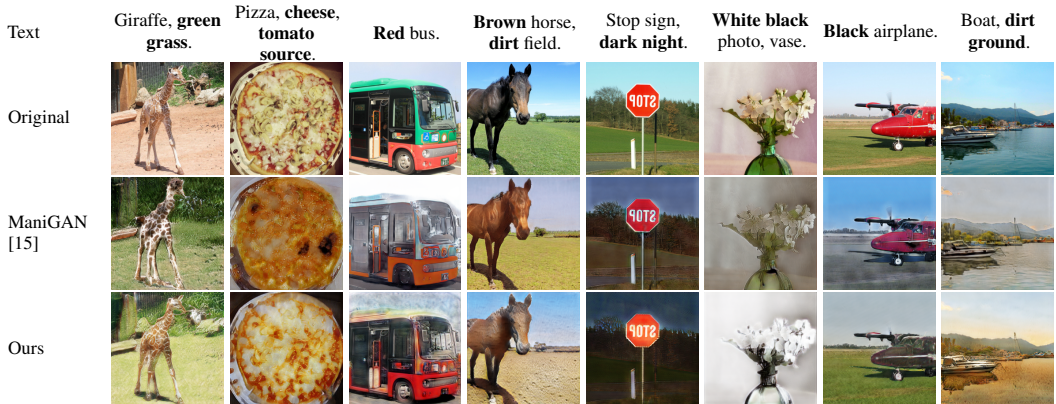

Figure 5: Qualitative comparison of two methods on the COCO dataset.

on CUB and 10 epochs on COCO using the Adam optimiser [12] with learning rate 0.0002. The hyperparameters $\lambda_1$, $\lambda_2$, $\lambda_3$, and $\lambda_4$ are all set to 1 for both datasets.

## 4.1 Comparison with State of the Art

**Quantitative comparison.** To evaluate the quality of synthetic images, the Fréchet inception distance (FID) [8] is adopted as a quantitative evaluation measure. Also, to better verify the manipulation performance of our method, we conducted a user study. For each dataset, we randomly selected 30 images with randomly chosen description (for COCO, each text-image pair is from the same category). Thus, there are in total 60 samples from two datasets for each method. Then, we asked workers to compare two results after looking at the input image, given text and outputs based on

two criteria: (1) accuracy: whether the modified visual attributes of the synthetic image are aligned with the given description, and text-irrelevant contents are preserved, and (2) realism: whether the modified image looks realistic. Finally, we collected 1380 results from 23 workers, shown in Table 1. Besides, to evaluate the efficiency, we record the runtime for training a single optimisation epoch (RPE) and the inference time (IT) for generating 100 new modified images. In the experiments, FID is evaluated on a large number of modified samples produced from mismatched pairs, i.e., randomly chosen input images edited by randomly selected text descriptions.

As shown in Table 1, compared with the state of the art, our method achieves better FID values on both CUB and COCO datasets, and both the accuracy and the realism show that our results are most preferred by workers. Besides, as shown in Table 2, the RPE and IT of our method are obviously smaller, and our method also has a much smaller number of parameters in both the generator and the discriminator when it has similar settings (e.g., size of hidden features) as the ManiGAN. Moreover, thanks to our powerful discriminator, we can even further reduce the number of parameters without sacrificing significant image quality. Table 2 "Ours*" records the smallest number of parameters that our model can have without significantly affecting the performance. Also, Fig. 3 plots the FID values corresponding to a different number of parameters in the generator and the discriminator.

This indicates that (1) our method can produce high-quality modified results with good diversity, (2) synthetic images are highly aligned with the given descriptions, while effectively preserving other text-irrelevant contents, and (3) our method is more friendly to memory-limited devices without sacrificing image quality.

**Qualitative comparison.** Figs. 4 and 5 show visual comparisons between ManiGAN and our method on CUB and COCO. As we can see, ManiGAN fails to produce satisfactory modified results on both datasets, and text-irrelevant contents are also changed. For example, in Fig. 4, the ManiGAN colours the branch using the colour of bird, and in Fig. 5, it produces distorted objects (e.g., giraffe and pizza), and even fails to produce required attributes (e.g., colour of airplane and style of the vase). We think the less satisfactory performance of ManiGAN is mainly because the discriminator fails to provide fine-grained training feedback related to each word.

## 4.2 Component Analysis

**Effectiveness of the word-level discriminator.** To verify the effectiveness of our proposed word-level discriminator, we conducted an ablation study shown in the Fig. 6 (c). As we can see, without the proposed word-level discriminator, the model fails to achieve an effective manipulation on both datasets. For example, the model cannot produce a yellow head and a yellow belly of the bird, and branches are coloured red and black as well. This indicates that due to the lack of word-level training feedback, the generator fails to disentangle different visual attributes, and then cannot build an appropriate word-region connection to achieve an effective modification. Also, FID values shown in Table 1 further verify the effectiveness of our proposed discriminator, as the values worsen significantly when the model does not implement the proposed word-level discriminator.

**Comparison between different word-level discriminators.** To verify the superiority of our word-level discriminator, we also conducted a comparison study between the text-adaptive discriminator [19], the word-level discriminator introduced in ControlGAN [14], and our proposed word-level discriminator, shown in Fig. 6 (d) and (e). As we can see, the model with the text-adaptive discriminator (d) changes text-irrelevant contents (e.g., the background), which is mainly because of the loss of spatial information in the discriminator. Also, the model with the word-level discriminator introduced in the ControlGAN [14] (e) fails to produce required attributes (e.g., red bird without yellow parts and zebra without the blue sky), and some contents that are not described in the text are modified as well (e.g., the white and black branch), which is mainly caused by the coarse training feedback provided from the discriminator.

**Visualisation of attention maps.** To further verify the effectiveness of our proposed word-level discriminator, we visualise attention maps that are produced by models with/without different word-level discriminators, shown in Fig. 7. We can observe that the model with our proposed word-level discriminator can generate a better spatial attention map with a more accurate location, a finer shape aligned with objects, and a better semantic consistency between visual attributes and corresponding words. On the contrary, models with other two word-level discriminators fail to generate appropriate attention maps or only produce coarse attention maps in the wrong location.

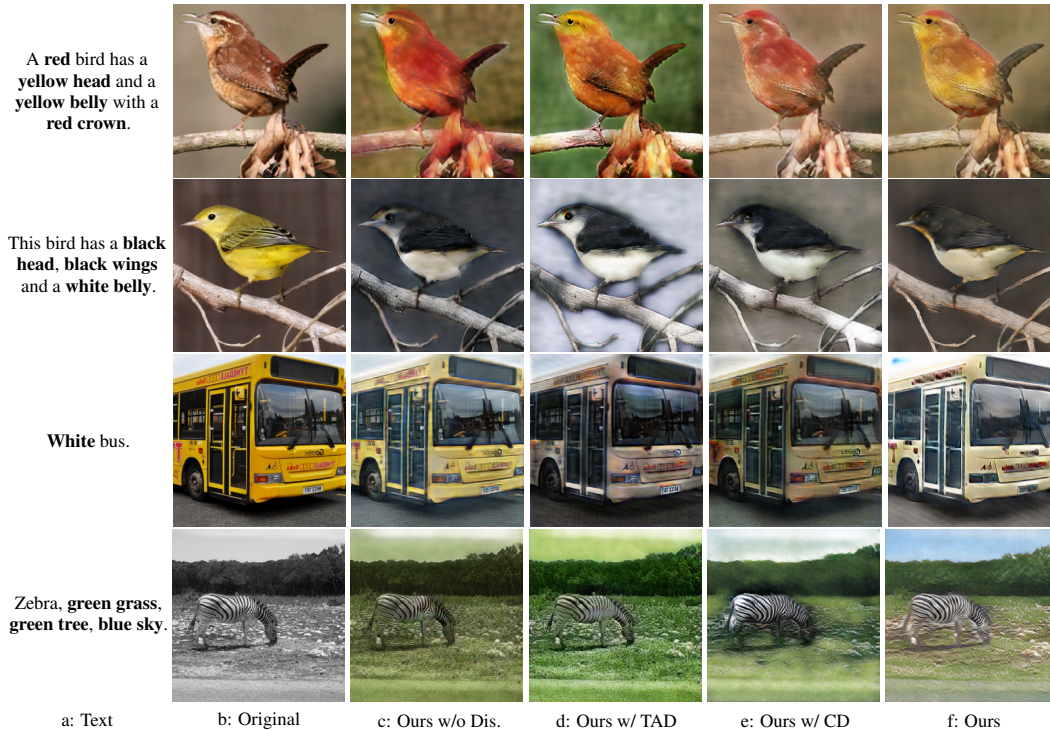

Figure 6: Ablation and comparison studies. a: text description with desired visual attributes; b: input image; c: removing the proposed word-level discriminator; d: replacing our discriminator with the text-adaptive discriminator [19]; e: replacing our discriminator with the word-level discriminator introduced in ControlGAN [14]; f: our full model.

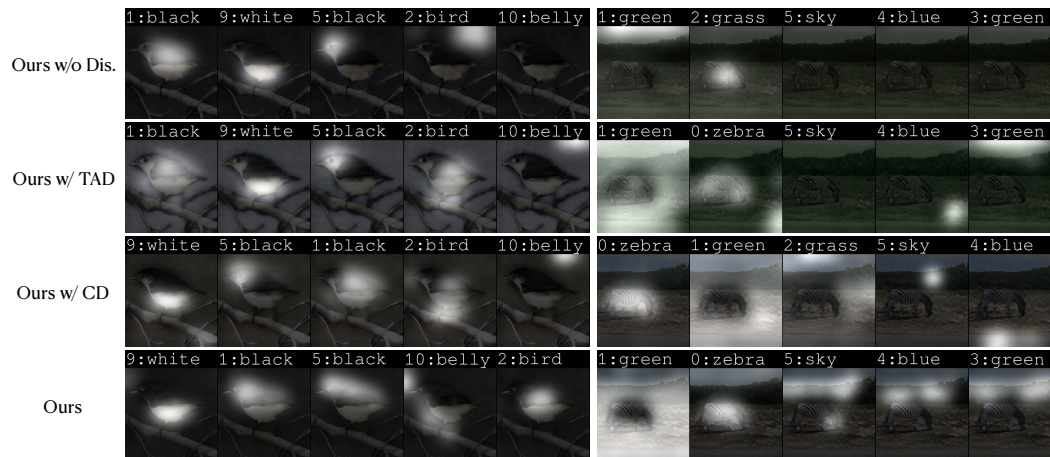

Figure 7: Visualisation of attention maps that are produced by models with/without different word-level discriminators.

## 5    Conclusion

We have proposed a novel and powerful word-level discriminator along with explicit word-level supervisory labels, which can provide the generator with fine-grained training feedback related to each word, and thus enable the construction of a lightweight architecture for image manipulation using natural language descriptions. More specifically, compared with the state of the art, our method has a much smaller number of parameters, but still achieves a competitive manipulation performance, which is friendly enough to memory-limited devices. Extensive experimental results demonstrate the efficiency and superiority of our method on two benchmark datasets.

## 6 Broader Impact

We have proposed a novel lightweight generative adversarial network for efficient image manipulation using natural language descriptions. To achieve this, we have introduced a powerful word-level discriminator, which can provide the generator with fine-grained training feedback in terms of each word in the given description, contributing to the construction of a lightweight architecture without sacrificing much image quality.

Furthermore, the proposed novel word-level discriminator can be easily implemented in other tasks involving different modality features, e.g., text-to-image generation and visual question answering, which is able to provide the generator with fine-grained supervisory feedback explicitly related to each word, to facilitate training a lightweight generator that has a small number of parameters, but can still achieve a satisfactory performance. Additionally, our powerful word-level discriminator allows to further simplify the architecture of the discriminator, enabling to build a complete lightweight network and thus providing the possibility to train a powerful and efficient neural network on memory-limited devices, such as mobile phones.

## Acknowledgments

This work was supported by the Alan Turing Institute under the UK EPSRC grant EP/N510129/1, the AXA Research Fund, the ERC grant ERC-2012-AdG 321162-HELIOS, EPSRC grant Seebibyte EP/M013774/1 and EPSRC/MURI grant EP/N019474/1. We would also like to acknowledge the Royal Academy of Engineering and FiveAI. Experiments for this work were conducted on servers provided by the Advanced Research Computing (ARC) cluster administered by the University of Oxford. We also acknowledge the use of the EPSRC-funded Tier 2 facility JADE (EP/P020275/1).

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
