[Supplementary Material]

# Supplementary Material

**Bowen Li[1], Xiaojuan Qi[2], Philip H. S. Torr[1], Thomas Lukasiewicz[1]**
[1]University of Oxford, [2]University of Hong Kong
{bowen.li, thomas.lukasiewicz}@cs.ox.ac.uk
xjqi@eee.hku.hk, philip.torr@eng.ox.ac.uk

## 1 Architecture

As shown in Fig. 3, the generator consists of a text encoder, image encoders, and a series of upsampling and residual blocks, where the text encoder is a pre-trained bidirectional RNN [2, 9], and the image encoders are pre-trained Inception-v3 [6] and VGG-16 [5] networks.

### 1.1 Residual Block

As shown in Fig. 1, each residual block consists of two $3 \times 3$ convolution layers, two instance normalisations (INs) [7], and one GLU [1] non-linear activation function.

Figure 1: Architecture of the residual block.

### 1.2 Upsampling Block

As shown in Fig. 2, each upsampling block consists of one upsample function with nearest mode, one instance normalisation (IN), one $3 \times 3$ convolution layer, and one GLU non-linear activation function.

Figure 2: Architecture of the upsampling block.

architecture of our proposed lightweight network

Figure 3: Architecture of our model.

## 1.3 Trend of Manipulation Results

Following [3], we use paired data $(I, S) \rightarrow I$ to train our model, where S is the text description matching the image $I$. Therefore, there is a trade-off between the reconstruction of the original contents existing in the input images and the generation of new attributes aligned with the given text descriptions. To verify this trade-off, we investigate the change of the manipulation results when the training epoch increases. As shown in Figs. 4 and 5, we can easily observe that the visual attributes of the input images are modified smoothly, matching the given text descriptions, e.g., blue head, black eyerings, and red belly in Fig. 4, and green grass background in Fig. 5. However, when the epoch increases further, new modified attributes are gradually replaced by the original contents in the input image, and finally the synthetic images become almost the same as the input images.

Figure 4: Trend of the manipulation results over epoch increases on the CUB dataset.

Figure 5: Trend of the manipulation results over epoch increases on the COCO dataset.

## 2 Additional Results

Fig. 6 shows various colour manipulations on the same images. In Figs. 7 and 8, we show additional comparison results between our method and ManiGAN [3] on the CUB [8] and COCO [4] datasets.

Figure 6: Various colour manipulations on the same images.

This bird is **red** with a **red crown**, a **red head** and a **red belly**.

This **brown** bird has **wings** that are **brown**, with a **brown belly** and **black eyerings**.

This bird is **white and black** in colour, with a **white head** and a **black belly**.

This bird has a **white crown**, a **white head**, a **yellow beak**, and a **yellow belly**.

A small bird with a **yellow belly** and a **white crown**.

This red bird has **blue wings**, a **red head**, and a **red belly**.

| Given Text | Original | ManiGAN [3] | Ours |

Figure 7: Additional comparison results between ManiGAN [3] and Ours on the CUB bird dataset.

| Given Text | Original | ManiGAN [3] | Ours |
|---|---|---|---|

Zebra, **dirt**.

**White**, tram.

**White** boat, **green grass**, **grey sky**.

Pizza, **cheese**.

**Black** horse, **green grass**.

**Green**, **blue** bus.

Figure 8: Additional comparison results between ManiGAN [3] and Ours on the COCO dataset.