[Reviews · NeurIPS 2020]

Review 1

Summary and Contributions: This paper proposes two methods to improve the performance of the text-guided image-to-image translation model. 1. Adopt word-region-level discriminator and only consider words that represent visual properties. 2. Utilize only one pair of generator-discriminator to apply to memory-limited devices. The results of ablation study show that the proposed methods works as intended, but the technical novelty seems somewhat incremental.

Strengths: The authors clearly described why they proposed these methods, and support their claims through ablation studies. This work is related to the NeurIPS community, but I did not find major contributions.

Weaknesses: - The total training time is compared to the baseline to show that the proposed model is light. However, it is more convincing to compare the inference time or number of parameters of the generator. - The results preserve text conditions well, but it looks a bit unrealistic and blurry, especially on the COCO dataset. Human evaluation is required to claim the superiority of the results.

Correctness: The description of the method is clear, and it is demonstrated by experimental results.

Clarity: The paper is well written and easy to follow.

Relation to Prior Work: The limitations of previous works, and the differences between these and the proposed method are clearly explained.

Reproducibility: Yes

Additional Feedback: In relation to what I pointed out as weaknesses of this paper, I think that further analysis of the following can improve the paper. - Inference time comparison - Human evaluation on generated images ****************** POST REBUTTAL COMMENTS Through the authors’ rebuttal, the questions I presented were clearly addressed. In particular, I felt that the novelty of this model was a bit minor, but their explanation of why the smaller model was able to perform better than the existing models was reasonable and it changed my mind. In my opinion, rather than emphasizing that it is a lightweight model, it would be better to reveal in the title that the performance was improved through the word-level discriminator that gives a more explicit training signal. Since the benefits of the proposed methods are proven both quantitatively and qualitatively, I will raise my score from 4 to 6.


Review 2

Summary and Contributions: This paper proposes a method for text-to-image synthesis based on Generative Adversarial Network (GAN). The proposed method extends the model from [14] by introducing word-level alignment loss in discriminator and designing much light-weight generator that requires much less memory footprint. Experiments on two datasets of CUB bird and MS-COCO demonstrated the effectiveness of the proposed method compared to the previous state-of-the-arts.

Strengths: + The paper is generally well-written and easy to follow. + The generated image quality and speed seems to have a clear improvemnets over the existing state-of-the-arts.

Weaknesses: - The technical novelty of the proposed method is somewhat incremental since it is largely based on the work from [14] with some modifications to the generator and the discriminator architectures. The word-level training feedback in the discriminator seems to be the main technical contribution, but is not ground-breaking as it extends the auxiliary classifier in conditional GAN with multiple classes (i.e. pre-defined sets of words). - The approach to discover relevant and irrelevant parts of an image to the text description is very heuristic and may not be valid in general. Specifically, only the nouns and adjectives are chosen manually as text-relevant attributes, which convey a very limited context of general descriptions. Although it may allow a fine-control of the image content in a limited context, it reduces the capability of aligning rich context of the text to the image, often available in approaches learning to encode the whole sentence (e.g. [4]). Although authors made some justifications in Section 3.2.1 of using heuristic approach, it does not feel that this assumption holds in general. - It would be informative to include more baselines in the experiment. Current comparisons are mostly focused on ManiGAN. Also, it would be informative to include user study as there are no comprehensive metrics that measure the alignment and synthesis quality.

Correctness: I did not find any tehchnical or factual error in the proposed method.

Clarity: The details on experiments and models are missing in the main draft, which makes it less self-contained. For instance, how do you create a training data for text-guided image manipulation without actual manipulated ground-truth? What is L_{DAMSM} in Ln 160? Are two discriminators in Eq.(6) share the parameters? What is manipulative precision?

Relation to Prior Work: It would be inforamtive to highlight the limitation of using heuristics for word-level alignment. The approaches based on whole-sentence encoding may enjoy encoding much general and flexible context in the text compared to the proposed method.

Reproducibility: Yes

Additional Feedback: Please address the concerns in the paper weakness section in rebuttal. ======= POST rebuttal ============ I appreciate authors for their efforts made in rebuttal. It addresses some of my major concerns, especially regarding the capability of modeling rich context in a sentence. I raised my score to 6.


Review 3

Summary and Contributions: This paper proposes a new lightweight method for image manipulation with text description. In general, the proposed architecture uses a single generator and a new word-level discriminator incorporating word labeling to focus on specific attributes to be manipulated. The authors evaluate the proposed method on CUB and COCO comparing with ManiGAN and provide promising quantitative and qualitative results with ablation.

Strengths: - Lightweight image manipulation is very important and challenging. - Word-level feedback discriminator is novel. - The authors conducted extensive experiments and the results seem to be promising in terms of quality and speed.

Weaknesses: - My main concern is "lightweight" term. If I understood correctly, "Lightweight" is a main contribution of this paper. and "lightweight" came from achieving competitive performance with a single generator (G) and discriminator (D) compared to ManiGAN using multi-scale G and D. But, this is not clarified in Section 3 and the authors presented the TPE and total training time only for "lightweight". I recommend that Section 3.1 has more description on how to achieve lightweight in details. - 'L' is used without definition. Maybe, L means the sequence length? In figure 2, notation definition will help readers to understand (e.g. v, w, L, ...) - Lightweight means both faster and smaller. It is required to add the model size comparison in Table 1. - Why SISGAN and TAGAN results are in Figure 1 only? - For most figures, the adjectives are mainly color-related. How are the results on other types of adjectives? Also, the results on the same input image but various colors can be helpful. - In the CUB result of Figure 6, why is the index-word pair of w/o Dis different from others? And what means the order of word-images?

Correctness: This methods seems to be correct.

Clarity: This paper is easy to follow. But, for IS and MP, "a large number of" (L187, P6) phrase is not specific.

Relation to Prior Work: There is no related work on lightweight GANs such as [Aguinaldo et al. 2019, Chen et al. 2020, Li et al. 2020]. [Aguinaldo et al. 2019] Aguinaldo et al. Compressing GANs using Knowledge Distillation. arXiv 2019. [Chen et al. 2020] Distilling portable Generative Adversarial Networks for Image Translation, AAAI 2020. [Li et al. 2020] GAN Compression: Efficient Architectures for Interactive Conditional GANs. CVPR 2020.

Reproducibility: Yes

Additional Feedback: - As textual information, noun and adjective are used only. The reason why this method focuses on them looks reasonable. Are there any results on other parts of speech? - What means single optimization epoch? After rebuttal: ==================================== I thank the authors for their great efforts. I carefully read the other reviewers' comments and author response. The authors alleviate most of my concerns. I think the main contribution is lightweight model for text-to-image manipulation and they showed promising results. This might lead to practical usages of the text-to-image synthesis models despite their incremental novelty. So, I decided to raise my score to 7.


Review 4

Summary and Contributions: This paper addresses text-guided image manipulation; given an original image and a text that describes the desired attributes such as texture, color, and background, the objective is to modify the original image to match the text. The authors propose an adversarial learning method with a novel word-level discriminator. Although the whole network architecture is lighter than the current state-of-the-art method having multiple discriminators, the proposed method achieves more accurate manipulation.

Strengths: - A challenging problem to manipulate an original image according to a text. - An interesting approach, paying attention to the use of word-level information in the text. - Throughout experiments on several datasets. Both accuracy of image manipulation and quickness of training are shown.

Weaknesses: - The authors should add FID score to evaluate the quality of manipulated images. The use of IS only is sometimes not so correlated to subjective evaluation. - In [14], there are only good results with images successfully manipulated by the text. It is not the fault of this paper, but the authors can show some failure cases for further improvement and attracting more researchers to address this task.

Correctness: This paper seems correct.

Clarity: This paper is clearly written.

Relation to Prior Work: The difference between this work and previous ones is discussed enough. In this paper, the proposed method also uses affine combination module, which is proposed in [14]. The overall network architecture, however, significantly different from that in [14]. Especially, instead of multiple discriminators in [14], word-level discriminator with word labeling is proposed. The improvement of training time and accuracy is quantitatively shown.

Reproducibility: Yes

Additional Feedback: The authors provide good references, but some of arXiv preprints can be updated. For example, [17] is accepted in ICLR 2016. Additionally, after the deadline of NeurIPS 2020, [14] is published in CVPR 2020. After reading all reviews and the responses from the authors, I have decided to maintain my first score. The authors have replied to my questions exactly.

[Author Response · NeurIPS 2020]

Table 1: Quantitative comparison: number of parameters in generator (NoP-G) and discriminator (NoP-D) (million), inference time for generating 100 new modified images (IT), accuracy (Acc), realism (Real), and FID. The numbers for Acc and Real indicate the average percentage of images favoured by users for the method. For Acc and Real, higher is better, for others, lower is better.

| | CUB | | | | | | COCO | | | | | |
|---|---|---|---|---|---|---|---|---|---|---|---|---|
| Method | NoP-G | NoP-D | IT (s) | Acc | Real | FID | NoP-G | NoP-D | IT (s) | Acc | Real | FID |
| ManiGA | 41.1M | 169.4M | 4.71 | 34.06 | 42.18 | 9.75 | 53.3M | 377.6M | 10.03 | 22.03 | 32.47 | 25.08 |
| Ours | 5.4M | 1.6M | 1.08 | 65.94 | 57.82 | 8.02 | 7.4M | 3.5M | 1.12 | 77.97 | 67.53 | 14.79 |

Figure 1: Top: qualitative comparison between our method, SISGAN [4] and TAGNA [18] on CUB (left) and COCO (right). Bottom: various colours on a same image by our method.

**R3: Comparison.** Please see Table 1 for inference time and number of parameters in generator and discriminator.

**R3: Blurry results on COCO.** Because our model has much fewer parameters, and generating realistic images on
COCO with text is more difficult. However, our method still outperforms ManiGAN both qualitatively and quantitatively.

**R3: User study.** Thanks for the suggestion. For each dataset, we randomly select 30 images with 1 randomly chosen
description (for COCO, each text-image pair is from the same category). Thus, there are 60 samples in total from two
datasets for each method. Then, we ask workers to compare two results after looking at the input image, given text and
outputs based on two criteria: (1) accuracy: whether the visual attributes of the manipulated image match the text, and
the text-irrelevant contents are preserved, and (2) realism: whether the manipulated image looks realistic. Finally, we
collected 1380 results from 23 workers, shown in Table 1. For both the accuracy and the realism, our results are most
preferred by workers. Also, both IS (Table 1 of the paper) and FID further verify the better performance of our method.

**R3, R4: Technical novelty.** The architecture of our method is fundamentally different from ManiGAN, as ManiGAN
has two modules: the main module and DCM. Once the main module is optimised, ManiGAN sets it to eval mode and
trains the DCM subsequently. However, our method can be trained end-to-end with much fewer parameters (see Table
1). As for the proposed discriminator, it is not a simple extension of the auxiliary classifier, as we need to consider
how to combine cross-domain features together, and how to provide appropriate word-level target labels to calculate
word-level feedback. It is also much more accurate than other word-level discriminators [13, 18] discussed in Sec. 4.2.

**R4: Limited context of general descriptions.** Both generator and discriminator objectives (Eqs 5 and 6) have included
the full sentence ($S$) as conditional adversarial loss to convey rich context. Our word-level discriminator does not have
trainable weights, and just works as a new approach to calculate an auxiliary loss, which encourages the generator to
better disentangle different attributes, and actually may not affect the alignment of rich text context to the image.

**R4: More baselines.** Thanks for your suggestion; we will include more baselines in the paper. We only compare our
method with ManiGAN, because both SISGAN and TAGAN fail to achieve an effective manipulation on both datasets,
shown in Fig. 1 (top). As for the user study, please see above "R3: User study".

**R4: Clarity.** Following ManiGAN, there are no ground-truth modified images in our training. We just use paired
data $(I, S) \rightarrow I$ to train the model, and our model is required to jointly solve text-to-image generation ($S \rightarrow I$) and
text-irrelevant contents reconstruction ($I \rightarrow I$). $\mathcal{L}_{\text{DAMSM}}$ is referred to [28], and manipulative precision is to [14].

**R5: Lightweight.** Thanks for your suggestion. Saying "lightweight" means that our model has a much simpler
structure with fewer parameters, but it still achieves a competitive performance. One reason to achieve this is that
our word-level discriminator provides much accurate word-level training feedback related to each specific attributes,
helping the generator to recognise them easily and thus enabling the possibility to simplify the architecture. The better
disentanglement can be verified by the visualisation of attention in Fig. 6. We will include more details in our paper.

**R5: Notation meaning.** $L$ represents the number of words in the word features $w$. Single optimisation epoch means
the running time per epoch. We will clarify this in the paper and add corresponding notation definition in Fig. 2.

**R5: Model size comparison.** Please see Table 1: our model has fewer parameters for both generator and discriminator.

**R5: No SISGAN and TAGAN results.** In Fig. 1 (top), SISGAN and TAGAN fail to achieve an effective manipulation
on CUB, and cannot produce realistic images on COCO. Thus, we only compare our method with ManiGAN.

**R5: Other adjective types.** For CUB, most descriptions are colours of different parts of a bird. For COCO, captions
are mainly about "type of objects + type of locations". So, our method mainly focuses on modifying the colours of a
bird for CUB, and the background or global style for COCO. See Fig. 1 (bottom) for various colours on the same image.

**R5: CUB result of Fig. 6.** Thanks for pointing this out. It is a mistake that a different random seed is chosen for "Ours
w/o Dis."; we will correct it in the paper. There is no specific meaning for the order; we just present some words with
corresponding highlighted visual regions to compare the performance of different word-level discriminators.

**R5: Results on other parts of speech.** Our proposed discriminator is used to provide additional word-level training
feedback related to visual attributes, enabling a better disentanglement. We think there is no need to include other parts
of speech in the proposed discriminator, because (1) it may not improve the manipulation ability, and (2) objective
functions have included the full sentence ($S$) as conditional adversarial losses to convey rich context of text.

**R6: Other evaluation matrix.** Thanks for your suggestion. Please see Table 1, our method achieves better FID scores.

**R6: Further improvement.** Thanks for your suggestion. First, how to produce higher-quality results involving
cross-domain features on the complex COCO dataset is still a challenge. Also, how to achieve an effective geometric
translation (e.g., horse <-> car) by using language is the other direction of our future work.

[Meta-Review · NeurIPS 2020]

The paper proposes a novel text-guided image manipulation method by proposing word-level discriminator loss. The proposed method is faster and requires less memory compared to existing models, and the experimental results show improvements over the baseline method (MainGAN). The paper initially received mixed ratings but the concerns were addressed by the rebuttal and all reviewers converged in favor of acceptance. The authors should revise the paper reflecting the reviewers’ suggestions and as promised by the rebuttal. NOTE FROM PROGRAM CHAIRS: For the camera-ready version, please expand your broader impact statement to discuss the potential negative impacts of your work, such as forgery and deepfakes, as well as possible mitigations.